# The ‘Surprise’ Question in Haemato-Oncology: The Estimating Physician and Time to Death Reduce the Prognostic Uncertainty—An Observational Study

**DOI:** 10.3390/cancers17081326

**Published:** 2025-04-15

**Authors:** Christina Gerlach, Martin Weber, Irene Schmidtmann

**Affiliations:** 1Department of Palliative Medicine, Heidelberg University Hospital, 69120 Heidelberg, Germany; 2Interdisciplinary Department of Palliative Care, III. Medical Clinic & Polyclinic, University Medical Center Mainz, 55131 Mainz, Germany; maweber@uni-mainz.de; 3Institute of Medical Biostatistics Epidemiology and Informatics (IMBEI), University Medical Center Mainz, 55118 Mainz, Germany; irene.schmidtmann@unimedizin-mainz.de

**Keywords:** palliative care, integration of palliative care in haematology, prognosis, decision making, hematologic malignancies, ‘Surprise’ Question

## Abstract

The ‘Surprise’ Question, an intuitive instrument used to estimate a patient’s survival, has been tested in different entities and settings. Despite the limitations in its accuracy, it has been recommended for the early integration of palliative care because the tool is suitable for triggering reflection on the patient’s overall situation. For the first time, we validated the ‘Surprise’ Question in haemato-oncology outpatients. The test sensitivity was poor for haematological malignancies (0.23) compared to solid tumours (0.58), i.e., many haematology patients in their last year of life were not identified (77%), but the specificity and predictive value for the survivors were safe, preventing patients with a better prognosis from receiving palliative care too early (94%). Notably, the accuracy of the ‘Surprise’ Question depended on the estimating physician rather than on objective prognostic factors. The clinical practice of using the ‘Surprise’ Question for screening for patients with limited survival may miss a substantial proportion of patients with haematological malignancies who may benefit from palliative care. The physician factors associated with skilful prognostication should be further explored.

## 1. Introduction

Despite the progress in cancer care, 50% of cancer patients die [1]. However, due to the progress of cancer care, many of them survive with cancer for a longer time than in the past. Nonetheless, patients with haematological malignancies less often receive specialised palliative care compared to patients with solid tumours [2,3,4,5].

Palliative medicine is a concept of care for patients with incurable diseases. It is based on Cicely Saunders’ Total Pain concept, which embraces not only the physical dimension of suffering but also the social, spiritual, and psychological dimensions [6]. Thus, palliative care uses an interdisciplinary and multiprofessional approach that addresses patients, their relatives and other informal care givers [7].

The concept includes the end-of-life. Although all medical disciplines deal with it, palliative medicine is the discipline that is associated with death and dying according to the public perception. Nonetheless, patients and their relatives essentially benefit from early integrated palliative care in the months and years preceding death. But the explicit acceptance of reaching the end of life is a hallmark of palliative care, which may frequently lead to denial if it is offered early or in a context of uncertain predictability.

The latter is a frequent argument in relation to haematologic malignancies (HMs) compared to solid tumours. In terms of acute leukaemia, at its onset, some patients move between death and survival. In situations of relapse or chronic HM courses, the response to therapies is uncertain. Besides the uncertain predictability, many haematologists prefer to deliver palliative care on their own. Thus, it is important to develop best practice models to identify early enough haemato-oncology patients with complex symptoms or who are approaching the end of life to provide palliative care when it is appropriate. Clinical trials demonstrated that subgroups of haemato-oncology patients benefited from integrated palliative care [8,9]. In view of the limited data available and the high symptom burden, the updated ASCO guidelines ‘Palliative Care for Patients with Cancer’ recently considered haemato-oncology patients in a specific section, suggesting that palliative care should be offered broadly and calling for more research on palliative care for haemato-oncology patients and their families [10].

The ‘Surprise’ Question, ‘Would you be surprised if this patient died in the next 12 months?’, aims to improve prognostic awareness in physicians [11,12] and other healthcare professionals [13,14,15]. It is a simple and feasible instrument for the intuitive estimation of mortality in patients with different kinds of malignant and non-malignant, chronic as well as advanced, diseases and in different settings inside and outside hospitals, in primary care, nursing homes and hospices [11,12,13,14,15,16,17,18,19,20]. Two recent meta-analyses showed acceptable test sensitivity of 71.4% and 69%, and specificity of 74% and 69% [15,18]. However, little is known about the use of the ‘Surprise’ Question in haematological malignancies [21,22].

The aim of this study was to test the feasibility of the ‘Surprise’ Question in haemato-oncology as a facilitator of early integration of palliative care by the determination of the face validity, observation of the application process and assessment of the survival at 12 months. Exploration of the applying physicians’ experience with the instrument revealed the potential of the ‘Surprise’ Question to involve haemato-oncologists’ intuition in the patient assessment [21]. We provided our preliminary quantitative data for two meta-analyses in 2017 [23,24], but since then, the ‘Surprise’ Question was tested only in haemato-oncology patients requiring inpatient care [22]. Thus, we decided to revisit and publish our quantitative findings from 2014, because the accuracy of the ‘Surprise’ Question in identifying people with poor prognoses has been questioned [23,24,25] and there is a persistent lack of palliative mortality prediction tools in haematology [26,27,28]. Here, we report the test results concerning the ‘Surprise’ Question when applied in a haemato-oncology outpatient clinic.

## 2. Materials and Methods

We performed a prospective cohort study and descriptive analysis of the validity of the ‘Surprise’ Question with an integrated qualitative analysis of the experiences of the physicians who applied the instrument.

### 2.1. Participant Selection and Study Procedures

The setting was the haemato-oncology outpatient clinic of a university hospital in Germany with an average of 1100 patients each quarter. From 1 October through 31 December 2012, doctors where asked to answer the ‘Surprise’ Question, ‘Would I be surprised if this patient died in the next 12 months?’, with ‘No’ or ‘Yes’ for each cancer patient on their daily appointment list. After receiving information about the ‘Surprise’ Question and the study aim, the doctors received their list in the morning, noted their responses on the list at a time of their choice during the course of the day, and returned it at the end of the day. In case of the answer ‘No, I would not be surprised’, physicians were further asked to indicate if the expected survival time could be even less than 3 months. Patients who presented repeatedly were assessed at each visit. Reassessment for survival status at 3 and 12 months was performed via a chart review at 18 months after inclusion. Included in the analysis were patients with current malignant disease; the exclusion criteria were non-malignant haematological diseases and visits for diagnostic investigation with a benign result, e.g., lymphadenopathy. The appointment lists were checked on a daily basis and matched with the returned lists to avoid selection bias.

Sociodemographic data concerning the physicians were published in a corresponding qualitative study [21]. The sample was representative of the setting, with experienced and senior physicians as well as physicians in further training.

The quantitative part of this study was assessed as a quality improvement initiative (Ethics Committee of the Medical Council of the Province of Rhineland-Palatine, 19 September 2012).

### 2.2. Interpretation of the Test Accuracy

The terminology used for the test qualities is challenging. The ‘No’ response to the ‘Surprise’ Question presents a ‘positive’ response, which means an estimate of the patient possibly dying within 12 months, while ‘Yes’ means a prognostic estimate beyond one year. Sensitivity in this context refers to the probability of detecting patients by the ‘Surprise’ Question who will die within 1 year; specificity reflects the ability of the ‘Surprise’ Question to identify patients who will survive 12 months. The positive predictive value (PPV) reflects what happened, i.e., the proportion of patients who died within one year when the physician estimated ‘No, not surprised’; the negative predictive value (NPV) indicates the probability of a patient assessed as ‘Yes’ to survive 1 year. The area under the curve (AUROC) or c-statistics value compares the number of correct estimates (sensitivity) against the false (1–specificity). It helps to differentiate if physicians’ ‘Surprise’ Question estimate was better than the chance of recognising a patient nearing the end of life. A score of 0.5 means the ‘Surprise’ Question estimate was equal to the chance, while a score of 1.0 means that random coincidence seems to be excluded. The Cohen’s kappa describes the concordance between the physicians’ estimate and the actual outcome one year later [29]. The Kaplan–Meier curves describe the ability of the ‘Surprise’ Question to discriminate patients into two groups for the risk of death. The odds ratio (OR) describes the probability of how many times more frequently an event happened under a defined condition, e.g., patients dying in the ‘No’ group compared to the ‘Yes’ group.

### 2.3. Statistical Methods

The study population, with its demographic and clinical characteristics, was described with appropriate statistical parameters (e.g., numbers and frequencies for categorical variables, and means and standard deviations (SDs) for normally distributed variables). In order to compare the answer to the ‘Surprise’ Question and the actual outcome after 12 months, cross-tabulation was applied for the haematological and oncological diseases. The sensitivity, specificity, and positive and negative predictive values with exact 95% confidence intervals were computed. The Cohen’s kappa, c-statistic, and Pearson–Clopper confidence intervals with 95% confidence intervals were evaluated. To compare the survival experience between patients with an assumed life expectancy of up to 3 months, more than 3 months, up to 12 months and more than 12 months, we also computed the Kaplan–Meier estimates and performed a log-rank test. The c-statistics, which are identical to the area under the ROC curve, were compared using the DeLong test [30]. We used logistic regression to determine which factors could explain a realistic assessment of life expectancy. This was performed in a case-control view: ‘cases’ were patients for whom the outcome and previous assessment of life expectancy agreed; ‘controls’ were those with disagreement between the outcome and the previous assessment. The explanatory factors considered were death within 12 months of the initial estimate, time between initial estimate and death in case of death, physician in charge, age and gender of patient, type of disease (solid tumour vs. haemato-oncological disease) and intention of treatment (curative vs. palliative). Patients who were assessed by doctors who saw 3 patients or less and patients for whom the assessing physician was not documented were excluded from this analysis.

We used SAS version 9.4 and SPSS version 23, and the R package pROC was used to perform the DeLong test [30] and to calculate the confidence intervals for the c-statistics.

## 3. Results

A total of 864 patients from the outpatient clinics for haematology and oncology were assessed for eligibility within 3 months. Thirty-six patients were not screened with the ‘Surprise’ Question by their doctors for organisational reasons, while 148 patients were excluded because of a non-malignant diagnosis, resulting in 680 included for follow-up. Eight patients were lost for follow-up. Thus, 672 patients were eligible for analysis 3 and 12 months after the survival estimate by the ‘Surprise’ Question (Figure 1).

Most patients were diagnosed with haematological malignancy (n = 512; 76.2%). These patients had a mean age of 63 years (SD ± 14, range 19–90), and 59.4% were men. A total of 160 patients suffered from solid tumours, mainly lung cancer (43.1%), where 65% were men and they had a mean age of 61 (SD ± 15, range 14–89). A metastatic stage was present in 48% of patients with solid tumours; 65% of patients with multiple myeloma were in the advanced stage Salmon and Durie III. For patients with lymphoma, the Ann Arbour stage was consistently used, of which 57% were Ann Arbour I–III. No stage was documented in 8%, so performance (Karnofsky, ECOG) was inconstantly documented. More than two-thirds of the patients had palliative goals of care. The demographic and clinical characteristics of the participants are shown in Table 1.

### 3.1. Survival of Patients Stratified by Answer to the ‘Surprise’ Question

The median survival for those with an assumed life expectancy of up to 3 months was 113 days, while it was 459 days for those with an assumed life expectancy of more than 3 months and up to 12 months, and for those with an assumed life expectancy of more than 12 months, the median could not be observed, i.e., it was greater than 546 days. The log-rank test indicated different survival experiences between the groups (*p* < 0.0001). Patients with haematological malignancies in the ‘No’ group were 5.7 times (95% CI: 3.0; 10.8) more likely to die within one year compared to those in the ‘Yes’ group, and this was also true for patients with solid tumours, standing at 4.3 times (95% CI: 2.5; 7.5) (Table 2; Figure 2). Overall, patients with haematological malignancies had a higher survival probability, with a 1-year survival probability of 90.0%, compared to patients with solid tumours, who had a 1-year survival probability of 67.5% (*p* < 0.001) (Appendix A).

### 3.2. The ‘Surprise’ Question as a Diagnostic Tool to Identify Patients with Limited Life Expectancy

We compared the mortality of patients for whom the ‘Surprise’ Question had a positive answer, i.e., the treating physician would not be surprised if the patient died within 12 months (‘No’ group), with those patients for whom the ‘Surprise’ Question had a negative answer (‘Yes’ group). Considerably less haematology patients (n = 31, 6.1%) were assessed for the ‘No’ group compared with oncology patients (n = 50, 31.3%).

Of the 81 patients who were deemed to have a life expectancy not beyond 12 months, 42 (52%) died within 12 months (positive predictive value (PPV) 0.52, 95% CI: [0.40; 0.63]), 12 out of 31 haematology patients (PPV 0.39) and 30/50 oncology patients (PPV 0.60). In contrast, 591 patients were expected to survive beyond 12 months, of whom 528 survived (89%, negative predictive value (NPV) 0.89, 95% CI: [0.87; 0.92]), 0.91 for haematology and 0.80 for oncology patients. The NPV, indicating the probability of a patient assessed as ‘Yes’ in terms of survival for 1 year, was higher than the PPV in both groups, which can be defined as an implicit safeguard to prevent people with higher life expectancy from receiving palliative care too early (Table 2). However, many more haematology patients than oncology patients died despite being estimated to survive one year (false negative 77%, 41/53 vs. 42%, 22/52).

When comparing the answer to the ‘Surprise’ Question between the 105 patients who died within 12 months and the 567 patients who survived, we found a sensitivity of 0.40 (95% CI: [0.31; 0.50]) and a specificity of 0.93 (95% CI: [0.91; 0.95]). The sensitivity was 0.23 in patients with haematological malignancies and 0.58 in patients with solid tumours. The lower sensitivity in the group of haematological patients also reflects the physicians’ tendency to answer ‘Yes, surprised if this patient died’ (94%) (Table 2). We observed a trend of improved sensitivity and PPV when comparing the first and last estimates when patients were assessed repeatedly. This might reflect the decreasing interval to the actual time of death, or the increasing hits known to arise from repeated testing (Table 2).

The c-statistic value was 0.59 (95% CI: [0.53; 0.65]) for the estimates in haematology and 0.70 (95% CI: [0.62; 0.77]) in solid tumours, indicating moderate accuracy. The Cohen’s kappa was 0.37 (95% CI: [0.27; 0.46]), indicating fair agreement between the physicians’ estimate and the actual outcome [29]. In summary, patients in the ‘No’ group had a many times greater risk of dying within 1 year compared to patients in the ‘Yes’ group. The accuracy of the survival estimate made with the ‘Surprise’ Question in terms of identifying patients at risk of dying in the next year was better in patients with solid tumours compared to those with haematological malignancies (*p* = 0.036).

### 3.3. Which Factors Contribute to a Realistic Assessment of Life Expectancy?

We fitted a logistic regression model with dependent variable agreement of assessment and outcome and explanatory variables:

death within 12 months of initial estimatetime between initial estimate and death in case of deathassessing physicianage and gender of patienttype of disease (solid tumour vs. haemato-oncological disease)intention of treatment (curative vs. palliative)

We found significant effects of death within 12 months of the initial estimate (*p* = 0.005), time between the initial estimate and death in the case of death (*p* < 0.001) and assessing physician (*p* = 0.0096) (Table 3). The age and gender of patients, type of disease and intention of treatment did not show an association with the agreement between the estimate and the outcome. For patients dying within 12 months, the odds ratio (OR) of having a realistic assessment was 0.012 (95% CI: [0.004; 0.036]), which means a realistic assessment was obtained more easily in patients who survived. The time between the initial assessment and death resulted in an OR of 0.991 per day (95% CI: [0.986; 0.996]), i.e., a realistic assessment was more likely in patients who died soon after the assessment.

There were some differences in the amount of agreement between the initial assessment and the outcome between physicians. While the ‘Surprise’ Question estimate by doctor 4 agreed with the outcome in 74/79 (94%) cases, the others provided less realistic assessments. The odds ratios compared to doctor 4 ranged between 0.117 (95% CI: [0.027; 0.502]) and 0.501 (95% CI: [0.097; 2.592]). The accuracy of the ‘Surprise’ Question estimate depended on the closeness to death and on the assessing physician rather than training while on study or the patients’ category of illness.

## 4. Discussion

This is the first study to examine the ‘Surprise’ Question, ‘Would I be surprised if this patient died in the next 12 months?’, in haemato-oncology outpatients. We observed the use of the ‘Surprise’ Question in outpatient settings at a university hospital and compared the estimates for haematology patients with those for patients with solid tumours. We found that the ‘Surprise’ Question answers discriminate well between patients at higher (‘No’) vs. lower (‘Yes’) risk of 1-year mortality, as others did before us [12,16,23,24,31,32,33,34]. The test accuracy was better in oncology compared to haematology patients and increased between the first and last ‘Surprise’ Question estimate, but this effect was attributable to the assessing physician and the proximity to death rather than the disease category.

The application of the ‘Surprise’ Question was feasible, but its usability for the integration of palliative care is ambiguous. The study conditions facilitated regular reflection about individual patient survival, which may not be the case in standard clinical operation mode. Further, the integrated interviews with the applying haemato-oncology specialists revealed a lack of consequences of a ‘No’ answer (indicating patients supposed to be in their last year of life), which undermines the function of the instrument when applied by physicians not trained in palliative care [21].

In the Gold Standard Framework and SPICT [35,36], the ‘Surprise’ Question is used to direct attention towards further prognostic, symptom, and psychosocial assessment in general practice. We and others combined the ‘Surprise’ Question as a screening instrument for further PC needs assessment in the haemato-oncology population, which revealed a substantial part of the patients had a severe unmet burden that could be alleviated by the integration of palliative care [22,37]. In a cohort of 101 haematology and 46 solid tumour inpatients, 68.7% were correctly estimated to die within 12 months, which was better than in our cohort (52%). In contrast to our results, their subgroup analysis showed no PPV difference between the groups (haematological malignancies: 69/101, 68.3% vs. solid tumours 32/46, 69.6%; *p* = 0.88) [22]. However, this result may be in concordance with our observation that the ‘Surprise’ Question estimates were more accurate when closer to death, because Hudson et al. observed hospitalised patients who might have been in a more severe status of illness than the outpatients of our cohort: 42.9% of their inpatient cohort died during the index and the following admission [22]. The uncertainty of the condition may also impact the test sensitivity. Van Lummel et al. [18] discovered in their meta-analysis of 88,000 assessments that the ‘Surprise’ Question had a worse sensitivity of 49.1% in the data from emergency departments, comparable to the worse test sensitivity of our haemato-oncology outpatients (23% and 38%), as opposed to oncology (83.8%) and non-malignant pulmonary diseases (82.5%). Nevertheless, an alternative reason for the improved test accuracy in our study close to the actual time of death may be the increasing hits known to arise from repeated testing or closer observation of the patient. Also, others found the better accuracy of the ‘Surprise’ Question estimates close to death. Gupta et al. [15] found in their meta-analysis of 56 studies with nearly 69,000 patients that the test accuracy of the ‘Surprise’ Question improved with the frequency of testing and a short time frame. They also described the positive impact of the physician’s experience on the ‘Surprise’ Question test accuracy, which we observed in our cohort. The risk that the ‘Surprise’ Question might reflect the predictive skills of the applying healthcare practitioner rather than the remaining lifespan was intensively discussed by Davis et al. [19], but the benefit of the ‘Surprise’ Question for haematology patients might be to inspire their physicians to think of and train in providing a prognosis [21].

The question is whether the exact prognosis is rather relevant in circumstances when the quality of life (QoL) is most relevant—even more when knowing the positive effect of specialist palliative care on QoL and in some cases even survival [38,39]. Nevertheless, the better accuracy of the ‘Surprise’ Question in identifying patients surviving than in those dying within 12 months may prevent too early end-of-life care in those patients, who may only need improved supportive care. However, palliative care should not be misinterpreted as end-of-life care only, which is an issue among haematologists, who are the first to inform their patients correctly about palliative care [5,40,41]. In this context, the interpretation of Mahes et al. [20] may be considered. Similarly to our results, they found the ‘Surprise’ Question to help identify patients with Parkinson’s disease surviving the next 12 months. Nevertheless, they interpreted that this function facilitates identification of patients with palliative care needs early enough to provide appropriate care in time.

The limitations of our study regarding generalisability and its relevance to current practice need to be discussed. This was a single-centre study in an academic hospital setting. Yet, oncology care is comparable in terms of the patient characteristics in outpatient settings between academic and non-academic settings and even between healthcare systems. However, the physicians’ characteristics may vary with greater continuity in patient care and a higher proportion of experienced staff in non-academic settings providing specialised haematology and oncology care. Our data are older, but not outdated. The ongoing debate on the integration of palliative care into haematology shows the continuing relevance of the topic [5,10,42,43,44,45]. Nevertheless, we hope for an improved understanding in future generations of specialists caring for patients with haematological malignancies in terms of palliative care, because it has been a mandatory subject in undergraduate medical education in Germany since 2013 [46,47].

## 5. Conclusions

As with any other instrument, the handling of the ‘Surprise’ Question needs training and practice. Our data suggest that the greatest accuracy of the survival estimation occurs when the string of intuition has a sounding board based on medical experience. The impact of the characteristics of the assessing physicians and the individual doctor–patient relation should be further explored to inform training on the ‘Surprise’ Question. However, the ability of the ‘Surprise’ Question to identify patients with longer survival may be an interesting approach to combine the activation of reflective intuition with the doctors’ natural preference for a positive prediction. Further, a tool does not only need a user but also a purpose. The impact of basic palliative care training and the availability of specialised palliative care services on the usability of the ‘Surprise’ Question needs to be examined. As the haematologists’ self-image embraces the provision of palliative care for patients and their relatives, haematologic malignancies should be considered with priority.

## Figures and Tables

**Figure 1 cancers-17-01326-f001:**
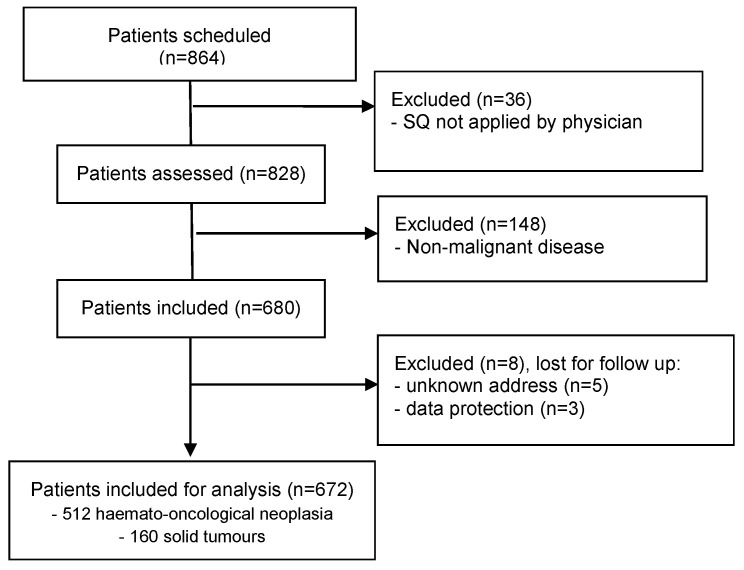
STROBE flowchart of patient recruitment, participation and analysis.

**Figure 2 cancers-17-01326-f002:**
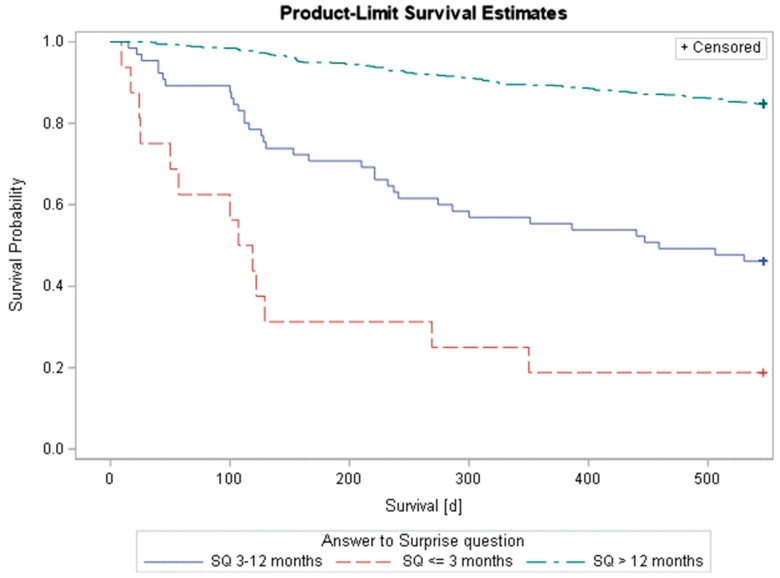
Kaplan–Meier survival function of patients stratified by the answer to the ‘Surprise’ Question.

**Table 1 cancers-17-01326-t001:** Demographic and clinical characteristics of the cohort.

Pts Characteristics	Haematol. ^1^ Pts, N = 512	Oncology Pts (N = 160)
Age in years, mean (range)	63 (19–90)	61 (14–89)
Female sex (%) Tumour site (%)	208 (40.6)	56 (35.0)
Lung cancer Sarcoma Renal cell carcinoma ENT carcinoma Prostate cancer Breast cancer Germinal cell tumour Urothelial cancer Hepatocellular cancer Medulloblastoma Mesothelioma Melanoma Colorectal carcinoma GIST High-grade NHL Multiple myeloma Low-grade NHL AML/ALL CLL Hodgkin’s lymphoma CML Waldenstrom’s disease Other MPS ZNS lymphoma Malignant thymoma MDS Hairy cell leukaemia	113 (22.1) 113 (22.1) 99 (19.3) 51 (10.0) 50 (9.8) 30 (5.9) 26 (5.1) 10 (2.0) 6 (1.2) 5 (1.0) 4 (0.8) 3 (0.6) 2 (0.4)	69 (43.1) 27 (16.9) 19 (11.9) 13 (8.1) 7 (4.4) 5 (3.1) 4 (2.5) 4 (2.5) 3 (1.9) 3 (1.9) 2 (1.3) 2 (1.3) 1 (0.6) 1 (0.6)

^1^ Haematol. = haematology; pts = patients; MPS = myeloproliferative syndromes.

**Table 2 cancers-17-01326-t002:** Test accuracy of the first and last estimate of survival with the ‘Surprise’ Question (1.0 meaning 100%).

	All (First Estimate)	All (Last Estimate)	Haematology (First Estimate)	Haematology (Last Estimate)	Solid Tumours (First Estimate)	Solid Tumours (Last Estimate)
Sensitivity	0.40	0.55	0.23	0.38	0.58	0.72
Specificity	0.93	0.91	0.96	0.94	0.81	0.77
Positive predictive value	0.52	0.55	0.39	0.45	0.60	0.62
Negative predictive value	0.89	0.91	0.91	0.92	0.80	0.85
c-statistics (AUROC 95% CI)	0.67 (0.62; 0.71)	0.73 (0.68; 0.78)	0.59 (0.53; 0.65)	0.66 (0.59; 0.72)	0.70 (0.62; 0.77)	0.75 (0.68; 0.82)
HR (95% CI)	6.9 (4.7; 10.2)	8.5 (5.8; 10.4)	5.7 (3.0; 10.8)	7.6 (6.6; 13.1)	4.3 (2.5; 7.5)	5.5 (3.0; 9.9)

1.0 meaning 100%; AUROC = area under the curve, receiver operating characteristics; HR = hazard ratio; CI = confidence interval.

**Table 3 cancers-17-01326-t003:** Effect of factors contributing to the test accuracy of the ‘Surprise’ Question.

Effect	OR	95% CI	*p*
Death within 12 months of initial assessment	0.012	0.004 to 0.036	<0.005
Time between initial assessment and death (in case of death)	0.991	0.986 to 0.996	<0.001
Physician 4	1.0		0.0096
Physician 1 vs. 4	0.232	0.053 to 1.016	
Physician 2 vs. 4	0.243	0.054 to 1.090	
Physician 3 vs. 4	0.117	0.027 to 0.502	
Physician 5 vs. 4	0.463	0.108 to 1.987	
Physician 6 vs. 4	0.501	0.097 to 2.592	
Physician 8 vs. 4	0.138	0.012 to 1.538	
Patient age	0.984	0.962 to 1.006	0.1584
Gender	1.298	0.733 to 2.300	0.3712
Entity (solid tumour vs. haematological malignancy)	0.669	0.339 to 1.323	0.2485
Goals of care (palliative vs. curative)	0.866	0.438 to 1.712	0.6792

OR: odds ratio; 95% CI: 95% confidence interval (Wald).

## Data Availability

The raw data are available on request from the corresponding author.

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
