# Peer review of "The ‘Surprise’ Question in Haemato-Oncology: The Estimating Physician and Time to Death Reduce the Prognostic Uncertainty—An Observational Study"

_cancers, 2025, doi:10.3390/cancers17081326_

Round 1

Reviewer 1 Report

Comments and Suggestions for Authors

The study is well designed and has novelty by using a survival curve to follow hemato-oncology patients with different estimated life expectancy by surprise questions, as well as comparisons between hematologic and solid malignancies.
Major comment:
1.    The methodology and analyses are mostly well done and informative. 
2.    I suggest the authors may also compare the survival curves between hematologic and solid malignancies (presented in the supplement is ok).
3.    Result section, line 200-201: In the sentence, “The accuracy of survival estimate with the ‘Surprise Question’ was better in patients with solid tumours compared to those with haematological malignancies,” please specify the statistical method used to define “better.” Additionally, include the corresponding p-value to support this comparison. This statistics may also be added to the footnote of Table 2.
Minor comment:
1.    Upon reviewing the literature, I found that a previous qualitative study investigated this issue in a similar fashion [1]. Additionally, another study included follow-up after the use of the Surprise Question [2]. Therefore, I suggest that the authors avoid stating, “This is the first study to examine the ‘Surprise Question’ in haemato-oncology outpatients.”
2.    In the Discussion section, line 288: “Limitations of our study in terms of generalizability and currency need to be discussed.” The term “currency” may be misleading. I recommend revising this to: “The limitations of our study regarding generalizability and its relevance to current practice need to be discussed.”
3.    
References:
1.    Gerlach C, Goebel S, Weber S, Weber M, Sleeman KE. Space for intuition - the 'Surprise'-Question in haemato-oncology: Qualitative analysis of experiences and perceptions of haemato-oncologists. Palliat Med. 2019;33(5):531-540.
2.    Hudson KE, Wolf SP, Samsa GP, Kamal AH, Abernethy AP, LeBlanc TW. The Surprise Question and Identification of Palliative Care Needs among Hospitalized Patients with Advanced Hematologic or Solid Malignancies. J Palliat Med. 2018;21(6):789-795.

Reviewer 2 Report

Comments and Suggestions for Authors

The study is interesting, but I think the manuscript cannot be accepted in its present form.

I have detailed my comments below:

Introduction:

In the introduction, the authors describe especially palliative care. Previous publications in the field of Surprise Question for chronic patients should be included in the introduction. Several reviews on this topic have been published in the last few years.

Materials and Methods:

Treatment programs and survival times have changed since the study was conducted 13 years ago. Why did the authors wait so long to publish?

Was each patient assessed by their attending physician? What was the physicians' experience? Were they either trainee physicians or qualified physicians?

Was all medical data available to the doctor? At what time was the question asked? Was it immediately following the consultation?

Results:

There is a lack of information on BMI, disease stage, co-morbidities. Survival times may be related to these data.

Other:

The qualitative study protocol and the Ethics Committee of the Medical were approved in 2015. The study was conducted in 2012. How do the authors explain this inconsistency?

Round 2

Reviewer 1 Report

Comments and Suggestions for Authors

I appreciate your efforts in revision far more than my expectations. 

Reviewer 2 Report

Comments and Suggestions for Authors

The manuscript is acceptable in its current form. The authors have addressed all of the comments.